# Promotion of Wound Healing and Prevention of Frostbite Injury in Rat Skin by Exopolysaccharide from the Arctic Marine Bacterium *Polaribacter* sp. SM1127

**DOI:** 10.3390/md18010048

**Published:** 2020-01-11

**Authors:** Mei-Ling Sun, Fang Zhao, Xiu-Lan Chen, Xi-Ying Zhang, Yu-Zhong Zhang, Xiao-Yan Song, Cai-Yun Sun, Jie Yang

**Affiliations:** 1State Key Laboratory of Microbial Technology, Institute of Marine Science and Technology, Marine Biotechnology Research Center, Shandong University, Qingdao 266237, China; sunml1990@yeah.net (M.-L.S.); zfang992@163.com (F.Z.); cxl0423@sdu.edu.cn (X.-L.C.); zhangxiying@sdu.edu.cn (X.-Y.Z.); zhangyz@sdu.edu.cn (Y.-Z.Z.); xysong@sdu.edu.cn (X.-Y.S.); suncy@sdu.edu.cn (C.-Y.S.); 2Laboratory for Marine Biology and Biotechnology, Qingdao National Laboratory for Marine Science and Technology, Qingdao 266237, China; 3College of Marine Life Sciences, Institute for Advanced Ocean Study, Ocean University of China, Qingdao 266003, China

**Keywords:** marine bacterium SM1127, exopolysaccharides, cell migration, skin wound healing, frostbite injury prevention

## Abstract

Many marine microorganisms synthesize exopolysaccharides (EPSs), and some of these EPSs have been reported to have potential in different fields. However, the pharmaceutical potentials of marine EPSs are rarely reported. The EPS secreted by the Artic marine bacterium *Polaribacter* sp. SM1127 has good antioxidant activity, outstanding moisture-retention ability, and considerable protective property on human dermal fibroblasts (HDFs) at low temperature. Here, the effects of SM1127 EPS on skin wound healing and frostbite injury prevention were studied. Scratch wound assay showed that SM1127 EPS could stimulate the migration of HDFs. In the full-thickness cutaneous wound experiment of Sprague–Dawley (SD) rats, SM1127 EPS increased the wound healing rate and stimulated tissue repair detected by macroscopic observation and histologic examination, showing the ability of SM1127 EPS to promote skin wound healing. In the skin frostbite experiment of SD rats, pretreatment of rat skin with SM1127 EPS increased the rate of frostbite wound healing and promoted the repair of the injured skin significantly, indicating the good effect of SM1127 EPS on frostbite injury prevention. These results suggest the promising potential of SM1127 EPS in the pharmaceutical area to promote skin wound healing and prevent frostbite injury.

## 1. Introduction

It has been found that a number of marine microorganisms synthesize exopolysaccharides (EPSs) [1]. Marine microbial EPSs are secreted into the surrounding environment [2], which protect cells via stabilizing membrane structure against external environmental pressures and/or serve as carbon and energy reserves [3]. Most of marine microbial EPSs are heteropolysaccharides, which are composed of three or four different monosaccharides that may be pentoses, hexoses, amino sugars, or uronic acids and are arranged in groups of ten or less to form repeating units [1]. Many marine microbial EPSs with novel compositions, structures, and properties have been reported to have potential in different fields, including adhesives, textiles, food additives, oil recovery, metal removal in mining, and industrial waste treatments, and so on [4,5,6,7,8,9]. Several marine microbial EPSs have been found to have potential in the pharmaceutical field. A sulfated EPS produced by the marine bacterium *Pseudomonas* sp. WAK-1 has cytotoxic activity against human cancer cell lines, such as MT-4 [10]. An EPS secreted by *Vibrio diabolicus* isolated from a deep-sea hydrothermal vent can promote the restoration of bone integrity in a mouse model [11]. An EPS from marine filamentous fungus *Keissleriella* sp. YS4108 can protect the rat pheochromocytoma line PC12 cells from hydrogen peroxide-induced injury [12]. The EPS from the marine bacterium *Pseudomonas stuzeri* 273 combats bacterial biofilm-associated infection [13]. These studies show the medical potential of EPSs from marine microorganisms. However, few marine microbial EPSs have been applied in the clinic. Thus, it is necessary to further study the pharmaceutical applications of marine EPSs. 

Skin, the largest organ in human and animal bodies, mainly protects the body against toxins, microorganisms, and some other potential insults and continuously maintains life. Patients with deep skin injuries are vulnerable to infection and metabolic imbalances, which sometimes result in amputation or even death [14]. Skin wound healing remains a significant clinical problem. Different methods, including ointments, wound dressings, and transplantation of grafts, have been developed to repair skin wounds [15,16]. Because synthetic drugs have a high risk of adverse reactions, such as allergy and drug resistance, natural products are becoming increasingly important and strongly suggested as alternative medicines for wound healing. Natural polymers, including chitin, chitosan, alginate, and hyaluronic acid (HA), have been applied in the preparation of wound dressings [17,18,19,20]. However, although a number of EPSs from marine microorganisms have been studied, no marine EPS has been reported to be applied in wound healing.

When skin is exposed to cold temperatures, frostbite injury might occur, which is most commonly observed in the exposed peripheral parts of the body, including hands, feet, cheeks, ears, and nose. People in low-temperature environments, especially high altitude mountaineers, soldiers in frigid environments, and scientific investigators have a high risk of suffering cold-induced injury [21]. Aloe vera ointment, thrombolytics, and iloprost have been used as primary and adjunctive therapies in the treatment of frostbite injury [22]. In addition to the treatment of the wounds after frostbite, frostbite prevention by pretreatment is also important. However, products with the property of preventing frostbite are still rather fewer. It has been reported that treatment with poly-l-arginine before frostbite can decrease the loss of skin tissue and thus protect the mouse skin [23]. So far, no EPS has been reported to be able to prevent frostbite injury in skin. 

*Polaribacter* sp. SM1127 is a marine EPS-producing bacterium isolated from Arctic kelp [24,25]. Previous study showed that this strain can secret an EPS with distinctive glycosyl composition and linkages [24]. Glycosyl composition analysis showed that SM1127 EPS consists mostly of *N*-acetyl glucosamine, mannose, glucuronic acid, with moderate amounts of galactose and fucose and minor amounts of glucose and rhamnose [24]. Glycosyl linkage analysis showed that this EPS is mainly composed of 4-linked glucuronopyranose, 2-linked galactopyranose, terminally linked galactopyranose, 4-linked glucopyranose, terminally linked fucopyranose, and 2, 3-linked mannopyranose, which indicates SM1127 EPS is hyper-branched [24]. SM1127 EPS has good antioxidant activity and outstanding moisture-retention ability [24]. In addition, SM1127 EPS has considerable protective property on human dermal fibroblasts (HDFs) at low temperature [24]. Moreover, SM1127 EPS has no oral toxicity and is nonirritating to skin according to safety evaluation [24]. Due to these good properties of SM1127 EPS, it is necessary to further probe its pharmaceutical potential. In this study, a scratch wound assay was conducted to detect the migration speed of HDFs in the presence/absence of SM1127 EPS. Moreover, full-thickness cutaneous wound experiments and frostbite injury experiments were performed to investigate the effects of SM1127 EPS on rat skin wound healing and frostbite injury prevention.

## 2. Results

### 2.1. SM1127 EPS Promotes HDFs Migration In Vitro

Effect of SM1127 EPS on HDFs migration was determined with a scratch wound assay. The wound closure rate was calculated at 0, 16, 24, 40, and 48 h after scratching. As shown in Figure 1, the wound closure rate was gradually increased with culture time, and the rates in the presence of the EPS were significantly higher than that in the absence of the EPS (*p* < 0.05). Moreover, the rate increased with the EPS concentration. Wound closure rates of cells treated with 0.5 and 1 mg/mL of the EPS after 48 h were 89.81% ± 1.44% and 94.92% ± 3.55%, respectively, 1.34 and 1.42 folds of that without EPS treatment (66.97% ± 4.43%). These data indicate that cell migration into the wound area is promoted in the presence of SM1127 EPS.

### 2.2. SM1127 EPS Increases the Healing Rate of Full-Thickness Cutaneous Wound

We further observed the effect of SM1127 EPS on skin wound healing in Sprague–Dawley (SD) rats with a full-thickness cutaneous wound model. The images of skin wounds treated with different concentrations of SM1127 EPS at different time points are presented in Figure 2A. On the 4th and 8th day, no obvious difference was observed between the wounds of the EPS treated groups and those of the control group. On the 12th day, wounds treated with the EPS were smaller than those untreated with the EPS. The wound treated with 5 mg/mL EPS daily healed nearly completely, while a vivid wound was still shown in the control group. On the 16th day, there was no visible wound on the skin treated with 5 mg/mL EPS. These results clearly show that SM1127 EPS can promote rat skin wound healing.

We further analyzed the wound healing rates of the control group and the EPS treated groups. As shown in Figure 2B, wound healing rates of the EPS treated groups were higher than those of the control group, and a higher concentration of the EPS was more effective. On the 12th day, wound healing rate of the control group was 69.43% ± 3.06%, which were significantly increased to 83.93% ± 4.11% for the 1 mg/mL EPS treatment group (*p* < 0.01) and to 90.25% ± 2.52% for the 5 mg/mL EPS treatment group (*p* < 0.001). On the 16th day, the wound healing rate of the 5 mg/mL EPS treated group was nearly 100%, but that of the control group was only 88.72% ± 3.21%. These data further show the promotion effect of SM1127 EPS on rat skin wound healing.

### 2.3. SM1127 EPS Accelerates Tissue Regeneration of Full-Thickness Cutaneous Wound in Histological Examination

To examine the effect of SM1127 EPS on tissue regeneration of the injured skins histologically, sections of the skin surrounding the wound areas in the skin wound experiment were stained and observed. The histological micrographs were shown in Figure 3. On the 4th day, a large number of inflammatory cells were observed in all groups. However, the number of inflammatory cells in the EPS treated groups was much lower than those in the control group. On the 8th day, granulation tissue and vascular-like structures appeared in the dermal layer of the 5 mg/mL EPS treated group but did not appear in the control group. On the 12th day, granulation tissue was observed in all groups, but the number of fibroblasts formed in the granulation tissue of the EPS treated groups was higher than that of the control group. After 16 days, the regenerated skins of all groups had no inflammatory cell, but the numbers of fibroblasts and blood capillaries in the granulation tissue of the control group were lower than those of the EPS treated groups. The skin morphology and architecture of the 5 mg/mL EPS treated group was fully developed with well-differentiated epidermal and dermal layers and subcutaneous tissue, showing a similar structure to normal skin. In contrast, the skin morphology and architecture of the control group were incomplete. These results indicate that SM1127 EPS can accelerate tissue regeneration when applied to skin wounds.

### 2.4. SM1127 EPS Pretreatment Increases Wound Healing Rate of the *Frostbitten Skin*

To investigate the effect of SM1127 EPS pretreatment on skin frostbite injury, an SD rat skin frostbite injury model was constructed by using cold iron plates stuck tightly on rat skins. The rat skins were pretreated by PBS or 20 mg/mL SM1127 EPS before frostbitten. Figure 4A showed the representative images of the healing process of the skin frostbite wounds in 18 days. The wounds were shriveled, scabbed, desquamated, and healed with time. On the 14th day, the scab of the EPS pretreated group fell off, but that of the control group still remained on the skin. On the 18th day, the wound of the EPS pretreated group nearly healed, while a vivid wound was still observed in the control group. We also calculated the wound healing rate of skin frostbite. Although the healing rates of the control group and the EPS pretreated group were similar on the 3rd day, the EPS pretreated group retained a significantly higher healing rate than the control group since the 7th day (Figure 4B), leading to the faster healing of the frostbite wounds of the EPS pretreated group (Figure 4A). These results show that SM1127 EPS pretreatment can increase the wound healing rate of the frostbitten skin.

### 2.5. SM1127 EPS Pretreatment Facilitates Tissue Repair of the Frostbitten Skin in Histological Examination 

To examine the frostbitten skins histologically, sections of the full-thickness skins in the center of the wound areas were stained and observed (Figure 5). On the 3rd day, many inflammatory cells exhibited in the control group, but only a few inflammatory cells were observed in the EPS pretreated group, indicating that the inflammation reaction in the skins of the EPS pretreated group was weaker. On the 7th day, granulation tissue was observed in both groups, but less edema fluid was observed in the granulation tissue of the EPS pretreated group. On the 10th day, the granulation tissue in the EPS pretreated group was more mature than that in the control group, because microvessels were observed in the shallow layer of the granulation tissue of the EPS pretreated group, but not in that of the control group. On the 14th day, thick epidermis was observed in the EPS pretreated group, but no epidermis structure was shown in the control group, which indicated that the tissue of the frostbitten skins in the EPS pretreated group regenerated faster than that in the control group because the integrity of epidermis is an index for skin wound healing [26]. These results indicate that SM1127 EPS pretreatment is beneficial to tissue repair and wound healing of the frostbitten skin. 

## 3. Discussion

Wound healing involves the processes of cell migration, cell proliferation, and matrix deposition [27]. During wound healing, dermis restoration is important, which provides space for the regeneration of microvessels and for enhancing the adherence of new epidermis [28]. HDFs are primary cells of the dermis, and their motility is a prerequisite for proper skin wound repair because these cells must migrate to the site of skin damage to be recruited for tissue repair [29]. Scratch wound assay is a classic and commonly used method to study cell migration [30,31,32]. In this study, we found that SM1127 EPS can increase the migration of HDFs in vitro by a scratch wound assay, which implies the possibility of SM1127 EPS to accelerate skin wound healing. Thus, the effect of SM1127 EPS on rat skin wound healing was further investigated. As expected, in a full-thickness cutaneous wound experiment, the SM1127 EPS treated groups showed a significantly higher healing rate and more intact skin structures compared to the control group, demonstrating the promotion effect of SM1127 EPS on wound healing of rat skins. The good ability of SM1127 EPS to promote skin wound healing may be attributed to its properties. SM1127 EPS has an excellent moisture-retention ability, which is better than that of the functional biopolymers generally used in wound dressings, such as HA, chitosan, and sodium alginate [24], and thus can provide cells a moist environment to favor cell migration and wound healing [33]. In addition, infection and inflammation of wounds usually cause the generation of an excess of reactive oxygen species (ROS), which destroy DNA structure, oxidize proteins and lipids, and hinder wound healing [34]. SM1127 EPS has a good antioxidant activity for scavenging DPPH•, •OH, O_2_^−^• [24], which may help in reducing the ROS level during tissue regeneration and, therefore, improve wound healing. 

One function of the EPSs secreted by microorganisms is to protect microbial cells against extreme environments [35]. EPSs produced by microbes in a cold environment, especially in the Arctic and Antarctic regions, usually play a significant role in protecting microbial cells from freezing injury [36]. The EPSs secreted by the Antarctic bacterium *Pseudoalteromonas arctica* KOPRI 21,653 and by the Arctic bacterium *Pseudoalteromonas* sp. SM20310 can increase the survival ratio of *Escherichia coli* in freeze–thaw cycles [37,38]. The EPSs produced by sponge-associated Antarctic bacteria *Winogradskyella* sp. strains CAL384 and CAL396, *Colwellia* sp. strain GW185, and *Shewanella* sp. strain CAL606 can improve the freeze–thaw survival ratios of the cells of themselves [39]. Due to this property, EPSs can be used in cryopreservation techniques [40]. However, the antifreeze effect of microbial EPSs on animal cells is rarely reported. We previously found that the EPS produced by the Arctic bacterial strain SM1127 can improve the survival ratio of HDFs at low temperature, suggesting its antifreeze effect on HDFs [24]. In this study, we further investigated the effect of SM1127 EPS on preventing frostbite injury in rat skin. The result showed that the healing rate of the SM1127 EPS pretreated group was higher than that of the control group. In addition, the inflammation of frostbite wounds of the SM1127 EPS pretreated group was weaker than that of the control group, and tissue repair of frostbite wounds of SM1127 EPS pretreated group was faster than that of the control group. These results indicated that the skins of the SM1127 EPS pretreated group suffered weaker injuries than the skins of the control group. Therefore, SM1127 EPS can reduce the damage of low temperature, especially freezing temperature, on skin, and thus has a good effect on preventing skin frostbite, suggesting that SM1127 EPS may have the potential to be developed as a useful cryoprotectant on skin.

In summary, the results in this study indicate that SM1127 EPS has a significant promotion effect on rat skin wound healing and good prevention effect on rat skin frostbite injury, which represents the first report that a marine microbial EPS has these properties. These results suggest the good potential of SM1127 EPS in the pharmaceutical area to be used in promoting skin wound healing and preventing skin frostbite injury.

## 4. Materials and Methods

### 4.1. Preparation of SM1127 EPS

The EPS of strain SM1127 was prepared as previously described [24]. Briefly, *Polaribacter* sp. SM1127 (CCTCC M 2013437) was cultured in a fermentation medium (30 g/L glucose, 10 g/L peptone, 5 g/L yeast extract, and 30 g/L sea salt, pH 7.0) for 5 days at 15 °C and 200 rpm. Then the bacterial cells in the culture were removed by centrifugation, and two volumes of cold absolute alcohol were added into the supernatant to precipitate the EPS. Proteins were removed from the EPS by protease hydrolysis (15 units/mL) with compound proteinase (GOLD WHEAT, Shanghai, China). After lyophilization, crude SM1127 EPS was obtained. The crude EPS was further purified by anion-exchange chromatography using a DEAE-Sepharose Fast Flow (GE, Boston, MA, USA) column (1.6 × 25 cm) and then by gel-filtration chromatography using a Sepharose 4B (GE, Boston, MA, USA) column (1.6 × 95 cm). The purified EPS was lyophilized, which was named SM1127 EPS in this study. SM1127 EPS was characterized by ^1^H NMR and two-dimensional correlation spectroscopy (COSY), total correlation spectroscopy (TOCSY), and heteronuclear single quantum correlation spectroscopy (HSQC) on a Varian Inova-500 MHz Bruker NMR spectrometer at 70 °C (Spectra are shown in Appendix A).

### 4.2. In Vitro Migration of HDFs

To examine whether SM1127 EPS had the property of promoting cell migration, a scratch wound assay was conducted with the method of Rodriguez et al. [41]. Briefly, on the bottom surface of each well of 6-well plates, two parallel lines were drawn across the middle of well using a marker. The other two guidelines were scratched horizontally in a row through the middle of the well. The lines intersected at right angles in the middle of the well, and these were reference points for taking pictures.

HDFs (ATCC PCS-201-012, Manassas, VA, USA) were seeded in 6-well plates at 3 × 10^5^ cells/well in Roswell Park Memorial Institute (RPMI) 1640 medium (Gibco, New York, NY, USA) with 10% fetal bovine serum (FBS) (Gibco, New York, NY, USA), and cultured in a humidified atmosphere with 5% CO_2_ at 37 °C for 24 h to 80%–90% confluence. The monolayer of HDFs was scratched between the guidelines with a 200-µL standard sterile pipette tip. Cells were gently washed twice with the growth medium to remove cell debris and then incubated with different concentrations (0, 0.5, and 1 mg/mL) of SM1127 EPS. The middle of each well along guidelines was observed by a Nikon Eclipse TE300 inverted microscope (Tokyo, Japan) and photographed by a Nikon digital camera (Tokyo, Japan) to record the scratch wound at 0, 16, 24, 40, and 48 h. The wound region was determined by tracing the wound margin with a fine-resolution computer mouse. The wound area was calculated using ImageJ software (National Institutes of Health, Bethesda, MD, USA) according to the number of pixels in the wound region. The wound closure rate was calculated with the following formula:(1)$$\mathrm{Wound}\;\mathrm{closure}\;\mathrm{rate}=(1-\frac{\mathrm{unhealed}\;\mathrm{wound}\;\mathrm{area}\;\mathrm{at}\;\mathrm{different}\;\mathrm{time}}{\mathrm{initial}\;\mathrm{wound}\;\mathrm{area}})\times 100\%$$

### 4.3. Animals

Animal experiments were approved by the Experimental Animal Ethics Committee of Shandong University (SDU-YZ-201806), which met the European Convention for the Protection of Vertebrate Animals used for Experimental and Other Scientific Purposes (Strasbourg, France, 18.03.1986).

Thirty male SD rats, weighing 200–220 g, were purchased from the Experimental Animal Center of Shandong University (Jinan, China). The rats were healthy and free of *Mycoplasma pulmonis*, *Pseudomonas,* and common rat viruses. They were housed in a moderate-security barrier and maintained on a 12-h light/dark cycle, with regulated temperature and free food and water intake. 

### 4.4. Full-Thickness Cutaneous Wound Model and SM1127 EPS Treatment

Fifteen male SD rats were used to conduct the skin wound experiment [42]. As shown in Figure 6A, rats were fixed on a plank and anesthetized with 3% pentobarbital sodium (30 mg/kg). Hairs on the dorsum of the rats were shaved, and exposed skins were sterilized with iodine solution. Four full-thickness circular wounds with a size of 1.8 cm in diameter were created on the dorsum of each rat using a pair of sharp scissors and a scalpel. The four wounds were marked as W0, W1, W2, and W3 and daily treated with 500 µL of phosphate-buffered saline (PBS, 50 mM, pH 7.4, as control), 0.5 mg/mL SM1127 EPS, 1 mg/mL SM1127 EPS, and 5 mg/mL SM1127 EPS, respectively. The EPS was dissolved in PBS and filter-sterilized before used. Rats were housed individually to prevent damage to the wound sites. On the 4th, 8th, 12th, and 16th day, the wounds were photographed, and the rats were sacrificed for histological examination. 

### 4.5. Frostbite Injury Model and SM1127 EPS Pretreatment

Fifteen male SD rats were used to perform the skin frostbite experiment [43]. As shown in Figure 6B, the legs and front teeth of rats were tied up on a plank. All the rats were in narcotism by injecting 3% pentobarbital sodium (30 mg/kg) into the cavum abdominis. Hairs on the dorsum of the rats were shaved, and the skins were cleansed with a 70% isopropyl alcohol swab. Two circles with a size of 2.5 cm in diameter were drawn on the exposure skins, marked as F0 and F1. F0 was applied with 500 µL PBS (as control), and F1 was applied with 500 µL 20 mg/mL SM1127 EPS that was dissolved in PBS and filter-sterilized before used (Pre-experiment showed that the SM1127 EPS solution with a concentration more than 20 mg/mL is too thick to be applied on skin). After 1 h for absorption, a cooled round iron plate with a size of 2.5 cm in diameter, which was precooled in crushed dry ice (−78.5 °C) for 15 min, was immediately stuck tightly on F0 or F1 for 1 min, and then a new cooled plate was immediately placed in the same location. The exchange of cold plates occurred in less than 5 s to prevent thawing. The exchange was repeated three times, making a freeze time slightly longer than 3 min. After the operation, rats were given subcutaneous injections of buprenorphine (0.05 mg/kg) for analgesia and housed individually to prevent damage to the frostbite sites. On the 3rd, 7th, 10th, and 14th day, the wounds were photographed, and the rats were sacrificed for histological examination. 

### 4.6. Measurement of Wound Healing Rate

In the skin wound experiment, the wounds of fifteen animals were imaged by a digital camera and coded after 4, 8, 12, and 16 days. In the skin frostbite experiment, the frostbite locations of rat skins were imaged and coded after 3, 7, 10, and 14 days. Before photography, a black sheet with a circular hole (2.5 cm in diameter) was put on the wound to be used as a constant area for standardizing wound size, guaranteeing the accuracy of subsequent planimetric quantitative analysis. The wound surface region was determined by tracing the wound margin with a fine-resolution computer mouse. The number of pixels corresponding to a surface area measurement was calculated using ImageJ software (National Institutes of Health, Bethesda, MD, USA). Wound healing rate at each imaged time point was calculated with the following formula:(2)$$\mathrm{Wound}\;\mathrm{healing}\;\mathrm{rate}=(1-\frac{\mathrm{wound}\;\mathrm{area}\;\mathrm{on}\;N\mathrm{th}\;\mathrm{day}}{\mathrm{wound}\;\mathrm{area}\;\mathrm{on}\;1\mathrm{st}\;\mathrm{day}})\times 100\%$$

### 4.7. Histological Examination

In the skin wound experiment, on the 4th, 8th, 12th, and 16th day, three rats were randomly chosen and sacrificed by intraperitoneal injection of excess pentobarbital. The skins surrounding wound areas of these rats were immediately excised, fixed overnight in 4% paraformaldehyde in PBS, and embedded in paraffin. 

In the skin frostbite experiment, on the 3rd, 7th, 10th, and 14th day, three rats were randomly chosen and sacrificed. The full-thickness skins in the center of the wound areas were fixed in 4% paraformaldehyde immediately and embedded in paraffin.

Sections of the fixed and embedded tissue were cut and stained with hematoxylin and eosin and observed by using a Carl Zeiss microscope (Axio Imager. A2, Jena, Germany). 

### 4.8. Statistics analysis

Data are shown as mean ± SD. Two-tailed Student’s *t*-test was used to analyze the statistical difference between the EPS-treated group and the EPS-untreated group at the same time. *p* < 0.05 was considered statistically significant.

## Figures and Tables

**Figure 1 marinedrugs-18-00048-f001:**
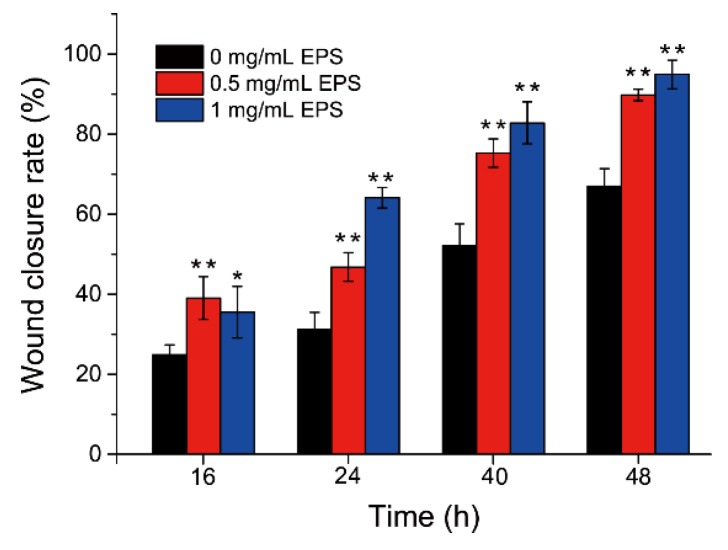
Effect of different concentrations of SM1127 exopolysaccharide (EPS) on the wound closure rate of human dermal fibroblasts (HDFs) in a scratch wound assay. The wound closure rate was calculated according to the unhealed wound area measured by using ImageJ software (National Institutes of Health, USA). * *p* < 0.05, ** *p* < 0.01.

**Figure 2 marinedrugs-18-00048-f002:**
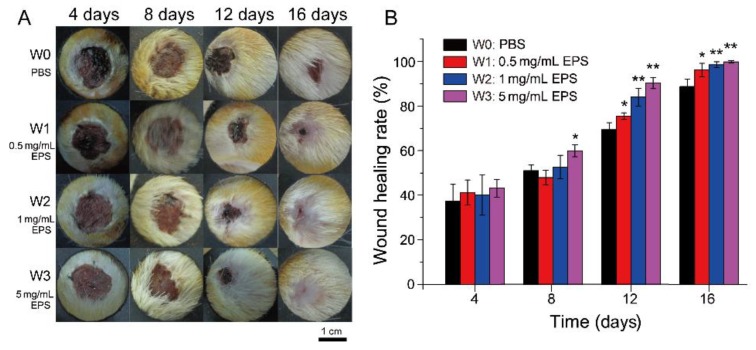
Effect of SM1127 EPS on full-thickness cutaneous wound repair of rat skins. (**A**) Representative images of skin wounds treated with 500 µL of PBS (W0, control), 0.5 mg/mL EPS (W1), 1 mg/mL EPS (W2), or 5 mg/mL EPS (W3) per day. (**B**) The healing rates of skin wounds treated with different concentrations of the EPS. * *p* < 0.05, ** *p* < 0.01.

**Figure 3 marinedrugs-18-00048-f003:**
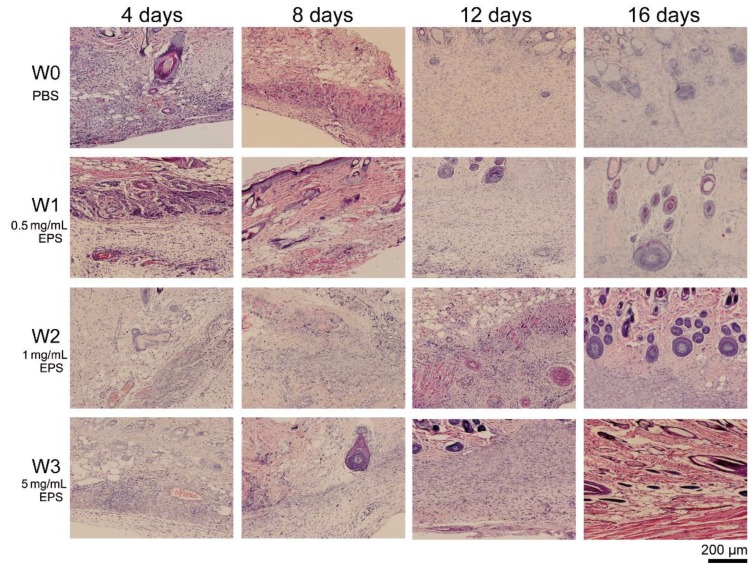
Histological images of rat skin wounds treated with 500 µL of PBS (W0, control), 0.5 mg/mL EPS (W1), 1 mg/mL EPS (W2), or 5 mg/mL EPS (W3) per day. Sections were stained with hematoxylin and eosin and observed by using a Carl Zeiss microscope (Axio Imager. A2, Germany).

**Figure 4 marinedrugs-18-00048-f004:**
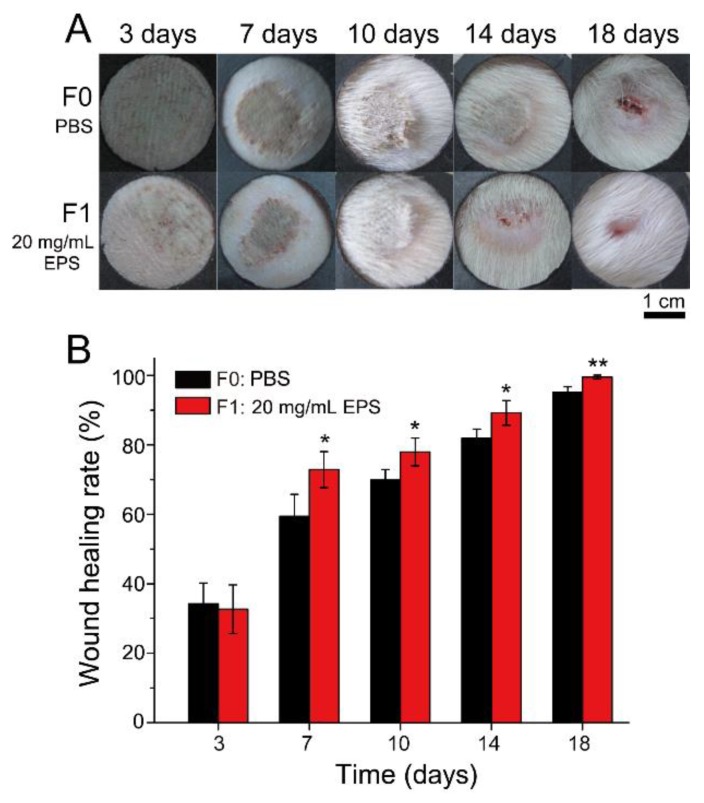
Effect of SM1127 EPS pretreatment on frostbite healing of rat skins. (**A**) Representative images of frostbitten wounds pretreated with 500 µL of PBS (F0, control) or 20 mg/mL EPS (F1). (**B**) The healing rates of frostbitten wounds pretreated with 500 µL of PBS or 20 mg/mL EPS. * *p* < 0.05, ** *p* < 0.01.

**Figure 5 marinedrugs-18-00048-f005:**
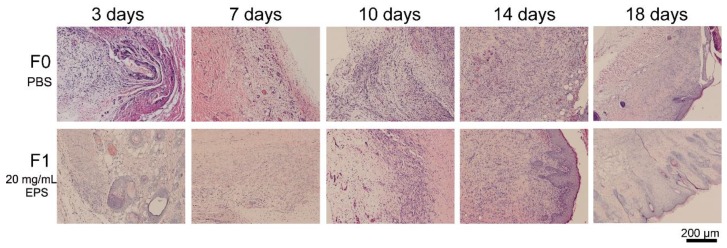
Histological images of cold-injured rat skins pretreated with 500 µL PBS (F0, control) or 500 µL 20 mg/mL EPS (F1). Sections were stained with hematoxylin and eosin and observed by using a Carl Zeiss microscope (Axio Imager. A2, Germany).

**Figure 6 marinedrugs-18-00048-f006:**
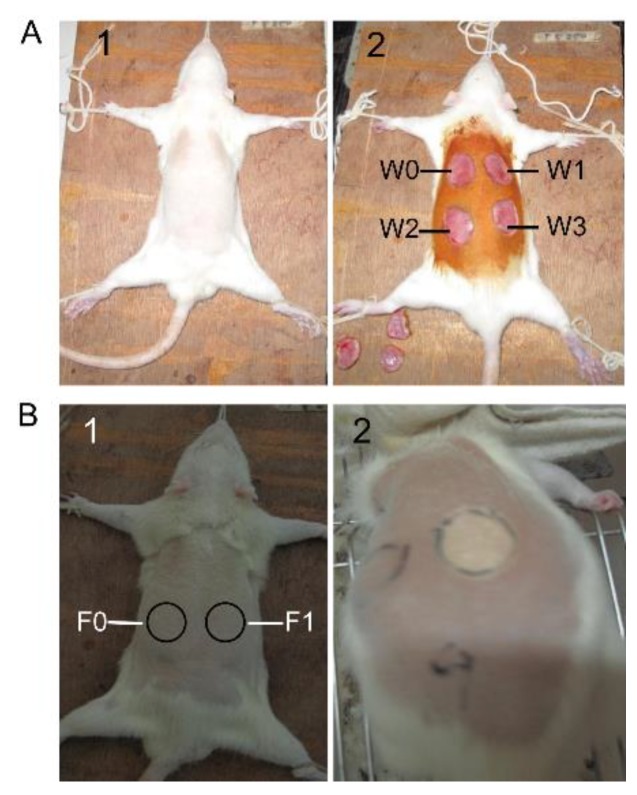
The processes of skin wounding (**A**) and frostbite injury (**B**) on the dorsum of SD rats. (**A1**) The rat was fixed on a plank, and hairs on the dorsum were shaved. (**A2**) Four full-thickness wounds with a size of 1.8 cm in diameter were created, marked as W0, W1, W2, and W3. (**B1**) Hairs on the dorsum of rats were removed, and two circles with a size of 2.5 cm in diameter were drawn, marked as F0 and F1. (**B2**) A frostbite injury was formed in F1 after cold plates were stuck tightly on it three times.

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
