# Peer review of "Promotion of Wound Healing and Prevention of Frostbite Injury in Rat Skin by Exopolysaccharide from the Arctic Marine Bacterium Polaribacter sp. SM1127"

_marinedrugs, 2020, doi:10.3390/md18010048_

Round 1

Reviewer 1 Report

The manuscript by Sun et al. described the use of exopolysaccharide from Polaribacter sp. SM1127 to treat the wound of rat skin. The authors suggested that SM1127 EPS showed promising properties for wound healing and forstbite prevention of skin and will be very useful for pharmaceutical area.The manuscript was very well written and easy to follow. The experimental design and results are convincing with proper statistical analyses.

Major comment

The authors mentioned about prevent frostbite injury, however based on section 2.4, it is more like the pre-treated skin with SM1127 EPS increases healing rate and not really prevent the frostbite. Maybe the authors can provide more explanation on this? Or modify the text? 

Minor comments:

Page 3 line 100: p < 0.005 can be removed (because it was 0.01 in Fig. 1).

Page 3 line 113-114: On the 16th day, there was no visible wound on the skin treated with 5 mg/mL EPS, but a small one still remained on the skin of the control group. - maybe just remove but a small one ..... (it makes a little confusion).

Page 7 line 188: Polaribacter sp. SM1127 instead of Strain SM1127.

Since the authors provide information about the structure of SM1127 EPS, would it be possible to provide its model structure (not necessary).

Author Response

Point 1: The authors mentioned about prevent frostbite injury, however based on section 2.4, it is more like the pre-treated skin with SM1127 EPS increases healing rate and not really prevent the frostbite. Maybe the authors can provide more explanation on this? Or modify the text?

 Response 1: Thank you very much for your constructive comments to our manuscript. In the frostbite experiment, SM1127 EPS was only applied on skin before frostbite. After frostbite, it could be seen from section 2.4 that the healing rate of SM1127 EPS pretreated group was higher than that of the control group, indicating the skins of SM1127 EPS pretreated group suffered weaker injuries than the skins of the control group. This was confirmed by the histological examination of section 2.5, in which, the inflammation of frostbite wounds of SM1127 EPS pretreated group was weaker than that of the control group and tissue repair of frostbite wounds of SM1127 EPS pretreated group was faster than that of the control group. All the results indicate that SM1127 EPS pretreatment can reduce skin damage, and thus has good effect on preventing frostbite injury.

To provide more explanation on this, we rewrote this part in the discussion (lines 238-253) as “The result showed that the healing rate of SM1127 EPS pretreated group was higher than that of the control group. In addition, the inflammation of frostbite wounds of SM1127 EPS pretreated group was weaker than that of the control group and tissue repair of frostbite wounds of SM1127 EPS pretreated group was faster than that of the control group. These results indicated that the skins of SM1127 EPS pretreated group suffered weaker injuries than the skins of the control group. Therefore, SM1127 EPS can reduce the damage of low temperature, especially freezing temperature, on skin, and thus has good effect on preventing skin frostbite, suggesting that SM1127 EPS may have potential to be developed as a useful cryoprotectant on skin.”

Point 2: Page 3 line 100: p < 0.005 can be removed (because it was 0.01 in Fig. 1).

Response 2: The indicated “p < 0.005” has been removed.

Point 3: Page 3 line 113-114: On the 16th day, there was no visible wound on the skin treated with 5 mg/mL EPS, but a small one still remained on the skin of the control group. - maybe just remove but a small one ..... (it makes a little confusion).

Response 3: “but a small one still remained on the skin of the control group” has been removed as suggested.

Point 4: Page 7 line 188: Polaribacter sp. SM1127 instead of Strain SM1127.

Response 4: The “Strain SM1127” has been replaced with “Polaribacter sp. SM1127”. Because the first paragraph of the discussion has been moved to the introduction according to the suggestion from another reviewer, this modification is not shown.

Point 5: Since the authors provide information about the structure of SM1127 EPS, would it be possible to provide its model structure (not necessary).

Response 5: The glycosyl composition and linkages, 1H NMR spectrum, two-dimensional COSY, TOCSY and HSQC spectra of SM1127 EPS have been obtained. According to these results, SM1127 EPS is hyper-branched and its structure is very complex, so its model structure is hard to be analyzed. For this reason, we could not obtain the model structure of SM1127 EPS.

Reviewer 2 Report

After having previously described the isolation, characterization and antioxidant activity evaluation of exopolysaccharide (EPS) from the Arctic Marine Bacterium Polaribacter sp. SM1127, in a continuation of their work, the authors describe in this manuscript, for the first time, the effects of SM1127 EPS on skin wound healing and frostbite injury prevention. The results justify the potential of SM1127 EPS in the pharmaceutical area to be used in promoting skin wound healing and preventing skin frostbite injury. The overall presentation of the manuscript is correct and it is well written.  I believe it will find a suitable interest and novelty for a wide number of readers of Marine Drugs. However, prior to publication, a minor revision is required as follows:

The bibliography is updated; nevertheless, some references are not the most appropriate and even unnecessary. My suggestion is to delete references 14,15, 23, 24. Considering Ref [26-29], Ref [36-37], Ref [40-41], Ref [42-44]: it is sufficient to put only one reference!

On the other hand, a reference should be after the following sentence: Line 37: “It has been found that a number of marine microorganisms synthesize exopolysaccharides  (EPSs).”

Line 63: put here the abbreviation of “hyaluronic acid” that appears in Line 213 as “HA”

Lines 87-91: Please delete this sentence because it is a repetition of what is said in the abstract and in the summary.

Line 150: “2.4. SM1127 EPS pretreatment increases wound healing rate of the frostbitten skin”. Authors should clarify why they chose 20 mg/mL as the concentration of SM1127 EPS to perform this study.

Line 246: “4.1. Preparation of SM1127 EPS”. At the end, it should be highlighted that the SM1127 EPS was characterized by 1H NMR, COSY, TOCSY and HSQC (spectra in the supplementary materials).

Lines 188-199: This whole paragraph should be moved to the introduction when describing SM1127 EPS and its properties.

Author Response

Point 1: The bibliography is updated; nevertheless, some references are not the most appropriate and even unnecessary. My suggestion is to delete references 14, 15, 23, 24. Considering Ref [26-29], Ref [36-37], Ref [40-41], Ref [42-44]: it is sufficient to put only one reference!

Response 1: According to your suggestion, the references 14, 15, 23 and 24 have been deleted. Ref [26-29], Ref [36-37], Ref [40-41] and Ref [42-44] have been put only one reference, respectively. Details are shown in the manuscript.

Point 2: On the other hand, a reference should be after the following sentence: Line 37: “It has been found that a number of marine microorganisms synthesize exopolysaccharides (EPSs).”

Response 2: A reference has been added after this sentence (line 38).

Point 3: Line 63: put here the abbreviation of “hyaluronic acid” that appears in Line 213 as “HA”

Response 3: The abbreviation “HA” has been added behind “hyaluronic acid” (line 63).

Point 4: Lines 87-91: Please delete this sentence because it is a repetition of what is said in the abstract and in the summary.

Response 4: The repetitive sentence has been deleted (lines 93-97).

Point 5: Line 150: “2.4. SM1127 EPS pretreatment increases wound healing rate of the frostbitten skin”. Authors should clarify why they chose 20 mg/mL as the concentration of SM1127 EPS to perform this study.

Response 5: In the section “4.5. Frostbite injury model and SM1127 EPS pretreatment” of the Materials and methods, the sentence “Pre-experiment showed that the SM1127 EPS solution with a concentration more than 20 mg/mL is too thick to be applied on skin” has been added to clarify why we chose 20 mg/mL as the concentration of SM1127 EPS to perform this study (lines 327-328).

Point 6: Line 246: “4.1. Preparation of SM1127 EPS”. At the end, it should be highlighted that the SM1127 EPS was characterized by 1H NMR, COSY, TOCSY and HSQC (spectra in the supplementary materials).

Response 6: At the end of section 4.1, the sentence about structural analysis “SM1127 EPS was characterized by 1H NMR and two-dimensional COSY, TOCSY and HSQC on a Varian Inova-500 MHz Bruker NMR spectrometer at 70°C (Spectra are shown in Supplementary Figures S1-S4)” has been added.

Point 7: Lines 188-199: This whole paragraph should be moved to the introduction when describing SM1127 EPS and its properties.

Response 7: The first paragraph of discussion has been moved to the introduction as suggested (lines 80-85).